# Clinical impact of ceruloplasmin levels at ANCA-associated vasculitis diagnosis

**Louis Camboulive**[1], **Frédérique Grandhomme**[2], **Nicolas Martin Silva**[1], **Kathy Khoy**[3], **Delphine Mariotte**[3], **Thierry Lobbedez**[4], **Anaël Dumont**[1], **Alexandre Nguyen**[1], **Hubert de Boysson**[1,5], **Achille Aouba**[1]*, **Samuel Deshayes**[1]

1 Department of Internal Medicine, CHU de Caen Normandie, Caen, France, 2 Laboratoire de Biochimie, CHU de Caen Normandie, Caen, France, 3 Laboratoire d'Immunologie et d'Histocompatibilité, CHU de Caen Normandie, Caen, France, 4 Department of Nephrology, CHU de Caen Normandie, Caen, France, 5 Normandie Univ, UNICAEN, UR4650 PSIR, CHU de Caen Normandie, Caen, France

* aouba-a@chu-caen.fr

**Data Availability Statement:** All relevant data are within the manuscript and its Supporting Information files.

## Abstract

### Objectives

Ceruloplasmin is an inhibitor of myeloperoxidase (MPO) activity that plays an important role in the pathophysiology of anti-neutrophil cytoplasmic antibody-associated vasculitis (AAV). This study aimed to evaluate the prognostic impact of serum level of ceruloplasmin at diagnosis in patients with anti-MPO antibody-positive AAV.

### Methods

This retrospective monocentric study in Caen University Hospital involved all consecutive adult anti-MPO antibody-positive patients with microscopic polyangiitis or granulomatosis with polyangiitis, diagnosed between January 2010 and January 2022 with available serum sample at inclusion. Patients outcomes were analyzed from two subgroups constituted according to the median serum level of ceruloplasmin. The same analyses were then performed in anti-proteinase 3 (PR3) antibody-positive patients.

### Results

Within the 92 patients analyzed, 50 patients had anti-MPO antibodies with a median ceruloplasmin level of 0.44 [quartiles 1–3, 0.40–0.49] g/L and a median Birmingham Vasculitis Activity Score of 19 [14–22]. After a median follow-up period of 40 [22–86] months, 13 (26%) patients had died: 10 (40%) in the low ceruloplasmin group and 3 (12%) in the high ceruloplasmin group (p = 0.03), with a significantly worse survival rate in the low ceruloplasmin group (p = 0.021). No significant differences in relapse rate or renal failure was observed between the two groups. The same analyses performed in the group of AAV patients with anti-PR3 antibody did not show any differences.

### Conclusion

In anti-MPO AAV patients, serum level of ceruloplasmin at diagnosis seems to be associated with a significant impact on survival.

**Funding:** The author(s) received no specific funding for this work.

**Competing interests:** The authors have declared that no competing interests exist.

## Introduction

Anti-neutrophil cytoplasmic antibodies (ANCA)-associated vasculitis is a group of diseases that include granulomatosis with polyangiitis (GPA) and microscopic polyangiitis (MPA). To this day, our understanding of these diseases is still incomplete. The inflammatory lesions and tissue damage seen in ANCA-associated vasculitis include, among other mechanisms, the consequences of the release of enzymes from activated neutrophils, such as myeloperoxidase (MPO) and proteinase 3 (PR3) [1]. MPO is a key component of neutrophil granules, capable of producing hypochlorous acid (HClO), which contributes to tissue destruction and inhibits α1 anti-trypsin, a plasma inhibitor of PR3 [2]. In ANCA-associated vasculitis, MPO is primarily expressed on the surface of activated neutrophils, where anti-MPO antibodies bind to it, triggering the release of reactive oxygen species. This process leads to the formation of neutrophil extracellular traps (NETs), which have cytotoxic properties that cause inflammation and vascular injury [3]. Ceruloplasmin is an anti-enzyme protein that belong to the α2-globulin group that is produced in large quantities during the acute phase of an inflammatory reaction and plays a major role in iron metabolism through its oxidoreductase activity. It has a strong ability to bind to and to inhibit the pro-inflammatory effects of MPO. Inadequate inhibition of MPO by low ceruloplasmin levels may contribute to a more severe phenotype in ANCA-associated vasculitis by enhancing oxidative stress and tissue damage through increased MPO activity and interaction with ANCA anti-MPO [4]. By contrast to α1 anti-trypsin deficiency that is associated with ANCA-associated vasculitis [5], there are no documented cases of ANCA-associated vasculitis associated with hypoceruloplasminemia so far. There has also been no evidence of an association between serum ceruloplasmin levels and the severity of ANCA-associated vasculitis. Because of its ability to inhibit MPO activity, we hypothesized that a low serum ceruloplasmin level could be associated with more severe vasculitis and worse prognosis. The objective of this study was to determine whether serum ceruloplasmin level at diagnosis had a prognostic impact in patients with anti-MPO antibody-positive vasculitis.

## Materials and methods

### Patient's selection

We performed a single-center retrospective study at the Caen University Hospital. All consecutive patients, diagnosed between January 2010 and January 2022 with GPA or MPA with anti-MPO or anti-PR3 ANCA detected by an automated chemiluminescence immunoassay (Quanta Flash®, Inova Diagnostics, San Diego) and with available serum at diagnosis for ceruloplasmin dosage, were included. Patients were retrieved through the immunology laboratory database and the database of hospitalizations, called "Programme de Médicalisation des Systèmes" (PMSI) with the following codes, according to the International Classification of Diseases, 10th revision (ICD-10): M313 and M317. Clinical, biological, and patients' outcome data were collected through the computerized medical records.

ANCA-associated vasculitis patients included in this study met the criteria of the Chapel Hill Consensus Conference [6]. Patients with vasculitis were classified as GPA or MPA according to the European Medicines Agency's (EMA) algorithm [7], and, in a post-hoc analysis, according to the recently published DCVAS criteria [8, 9]. Vasculitis activity was determined by Birmingham vasculitis score (BVAS) version 3 [10]. Patients with eosinophilic granulomatosis with polyangiitis (EGPA) were excluded as well as those with known renal failure before the diagnosis of vasculitis and attributable to another cause, active neoplasia, or ongoing infection at the time of diagnosis.

This study was conducted in accordance with the good clinical practices and the principles of the Helsinki declaration. In accordance with the French public health code (Art. L1121, Art.

L1122) and the favorable opinion of the committee for the protection of individuals of Normandie (CLERS, n˚1947), formal patient consent was not required. A letter of information regarding the issues and characteristics of the study was sent to each patient. The serum library is registered with the French commission for liberty and information (CNIL) (DC-2008-559).

## Ceruloplasmin dosage and patient subgrouping

Serum ceruloplasmin levels were measured by nephelometry (Beckman Coulter®) on a sample taken before any immunosuppressive treatment (reference level: 0.15–0.50 g/L). Patients with anti-MPO ANCA-associated vasculitis were divided into two groups according to the median ceruloplasmin level. The same analyses were then performed in the group of anti-PR3-positive patients and in the groups of patients with a GPA or MPA phenotype, divided into two groups according to the median ceruloplasmin level.

## Study's definitions

Relapse was defined as a recurrence of vasculitis in any organ requiring a therapeutic change. Chronic end-stage renal disease was defined as an estimated glomerular filtration rate (eGFR) $<15$ ml/min/1.73 m$^2$ or onset of chronic dialysis or renal transplantation. Renal survival was defined by the absence of death or end-stage renal disease. Proteinuria was considered positive if $>0.3$ g/L or $>0.3$ g/24h or $>3$ mg/mmol of creatinine according to the available results. Patients were followed until death, loss to follow-up, or 31.01.2022. Patient data were censored at the last follow-up visit.

## Statistic parameters

Qualitative variables were reported as percentages and compared using χ2 or Fisher tests, according to expected frequencies. Quantitative variables were expressed as medians and quartiles 1 and 3 and analyzed using the nonparametric Mann-Whitney test. Survival curves were estimated with the Kaplan-Meier method, and the inter-group difference was assessed with the log-rank test. A p value $<0.05$ was considered statistically significant. Calculations were performed using R statistical software (version 4.1.3) [11], survival curves and ROC curves were plotted with GraphPad Prism 7 (GraphPad Software Inc.). The cut off levels of serum ceruloplasmin to discriminate survival was determined using the Youden index.

## Results

Among 122 patients with ANCA-associated vasculitis diagnosed between 2010 and 2022, 30 patients were excluded: for 24 patients, serum before therapeutic introduction was not available, 3 were EGPA, 3 had known renal failure before diagnosis of vasculitis, none had active neoplasia or infection at diagnosis. Among the remaining 92 patients, 50 patients with anti-MPO ANCA were included with a median age of 68 [64–73] years and 25 (50%) women (Table 1). The median ceruloplasmin level was 0.44 [0.40–0.49] g/L, allowing the patients to be divided into the "low ceruloplasmin" and "high ceruloplasmin" group. The clinical findings, the biological data (except for ceruloplasmin levels) and the therapies used were comparable in the two groups (Table 1).

After a median follow-up of 40 [22–86] months, 13 (26%) patients had died: 10 (40%) in the low ceruloplasmin group and 3 (12%) in the high ceruloplasmin group (p = 0.03). The causes of death in the low ceruloplasmin group were: infectious events (n = 4), vasculitis relapse (n = 1), thromboembolic events (n = 2, i.e., stroke or acute coronary syndrome) and unknown cause (n = 3). In the high ceruloplasmin group, the 3 deaths were related to infectious events (n = 1) and thromboembolic events (n = 2). Of note, there was no statistical

**Table 1. Characteristics of 50 patients with anti-MPO ANCA-associated vasculitis with ceruloplasmin level available at diagnosis.**

| Characteristics | (n = 50) | Low ceruloplasmin (n = 25) | High ceruloplasmin (n = 25) | P value |
|---|---|---|---|---|
| **Demographic data** | | | | |
| Age at diagnostic (years) | 68 [64–73] | 71 [66–74] | 67 [63–71] | 0.17 |
| Woman | 25 (50) | 12 (48) | 13 (52) | 0.78 |
| **Vasculitis type** | | | | |
| GPA | 5 (10) | 3 (12) | 2 (8) | 1 |
| MPA | 45 (90) | 22 (88) | 23 (92) | 1 |
| **BVAS** | 19 [14–22] | 19 [14–22] | 18 [14–21] | 0.50 |
| **Characteristics of vasculitis** | | | | |
| General symptoms | 41 (82) | 20 (80) | 21 (84) | 1 |
| Dermatological symptoms | 7 (14) | 4 (16) | 3 (12) | 1 |
| Pulmonary symptoms | 29 (58) | 13 (52) | 16 (64) | 0.40 |
| ENT symptoms | 13 (26) | 8 (32) | 5 (20) | 0.34 |
| Ophthalmological symptoms | 6 (12) | 4 (16) | 2 (8) | 0.67 |
| Abdominal symptoms | 1 (2) | 0 (0) | 1 (4) | 1 |
| Neurological symptoms | 12 (24) | 7 (28) | 5 (20) | 0.51 |
| Cardiological symptoms | 1 (2) | 0 (0) | 1 (4) | 1 |
| Renal symptoms | 42 (84) | 21 (84) | 21 (84) | 1 |
| **Biological data** | | | | |
| Hematuria | 40 (87) (n = 46) | 19 (79) (= 24) | 21 (95) (n = 22) | 0.19 |
| Proteinuria | 34 (69) (n = 49) | 18 (72) | 16 (67) (n = 24) | 0.69 |
| Creatinine level (µmol/L) | 254 [131–506] (n = 49) | 238 [161–484] (n = 24) | 290 [108–564] | 0.97 |
| CRP (mg/L) | 82 [17–158] (n = 48) | 62 [10–130] (n = 24) | 118 [42–170] (n = 24) | 0.09 |
| **Treatment** | | | | |
| Induction | 43 (86) | 21 (84) | 22 (88) | 1 |
| Cyclophosphamide | 35 (70) | 17 (68) | 18 (72) | 0.76 |
| Rituximab | 8 (16) | 4 (16) | 4 (16) | 1 |
| Maintenance | 38 (78) (n = 49) | 18 (75) (n = 24) | 20 (80) | 0.68 |
| Rituximab | 15 (31) | 7 (30) | 8 (32) | 0.83 |
| Azathioprine | 21 (47) | 9 (46) | 12 (48) | 0.46 |
| Methotrexate | 2 (5) | 2 (9) | 0 (0) | 1 |
| Mycophenolate mofetil | 1 (3) | 0 (0) | 1 (4) | 1 |
| Plasma exchanges | 11 (22) | 4 (16) | 7 (28) | 0.31 |
| Bolus glucocorticoids | 42 (84) | 20 (80) | 22 (88) | 0.71 |
| Vaccination against *Pneumococcus* | 13 (26) | 4 (16) | 9 (36) | 0.11 |
| Trimethoprim-sulfamethoxazole | 39 (78) | 19 (76) | 20 (80) | 0.74 |
| **Relapses** | 9 (18) | 3 (12) | 6 (24) | 0.47 |
| **Deaths** | 13 (26) | 10 (40) | 3 (12) | 0.03 |
| **Chronic end-stage renal disease** | 10 (20) | 5 (20) | 5 (20) | 1 |
| **Follow-up (months)** | 40 [22–86] | 36 [18–77] | 46 [23–87] | 0.17 |

Values are given as headcount (%) or median [quartile 1-quartile 3].

GPA: granulomatosis with polyangiitis; MPA: microscopic polyangiitis; MPO: myeloperoxidase; BVAS: Birmingham vasculitis activity score; ENT: ear, nose and throat; CRP: C-reactive protein.

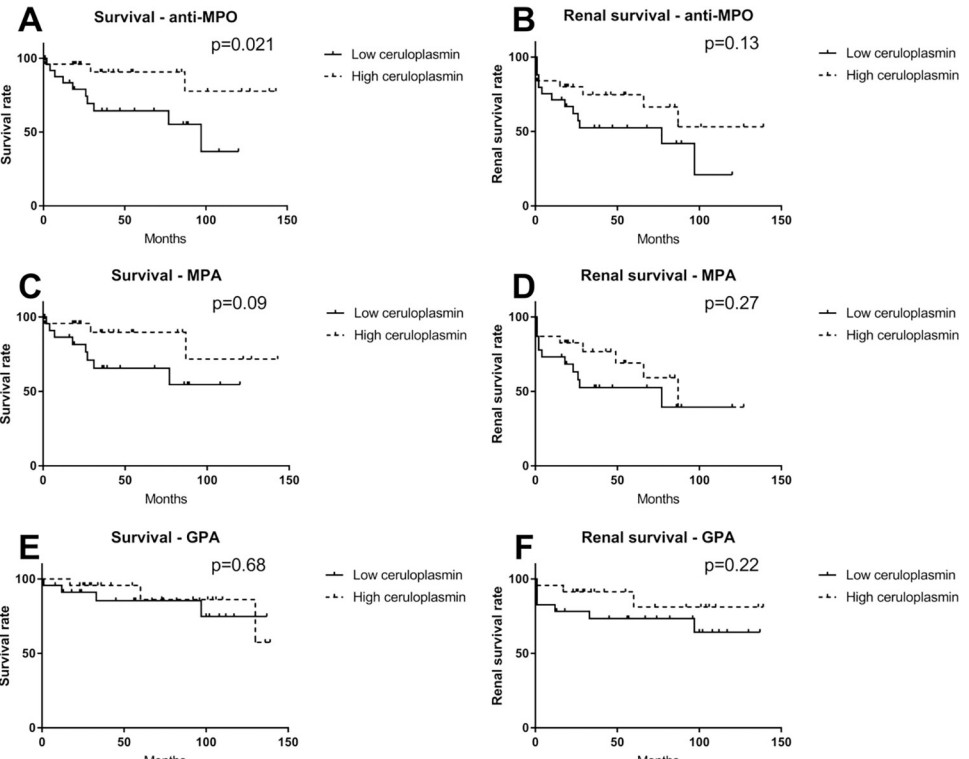

**Fig 1.** A. Survival curves of patients with anti-MPO ANCA vasculitis. B. Renal survival curves of patients with anti-PR3 ANCA vasculitis. C. Survival curves of patients with a microscopic polyangiitis phenotype. D. Renal survival curves of patients with a microscopic polyangiitis phenotype. E. Survival curves of patients with a polyangiitis granulomatosis phenotype. F. Renal survival curve of patients with a polyangiitis granulomatosis phenotype.

difference between the two subgroups concerning the BVAS score. There was no significant difference in the rate of chronic end-stage renal disease between the groups (p = 1). The relapse rate of vasculitis between the two groups was comparable (p = 0.47) (Table 1). We observed poorer survival in the low ceruloplasmin group compared with the high ceruloplasmin group (p = 0.021, Fig 1A). Renal survival was similar (p = 0.13, Fig 1B).

The cut off levels of serum ceruloplasmin to discriminate survival was 0.43. The ROC curve (S2 Fig) has an AUC of 0.65 (IC95% 0.49–0.81), for a sensibilty of 0.77 (IC95% 0.46–0.95), specificity of 0.59 (IC95% 0.42–0.75), positive predictive value of 1.9 (IC95%: 1.2–3.1) and negative predictive value of 0.39 (IC95%: 0.14–1.1).

Regarding antiinfection prophylaxis measures, both subgroups were comparable, 13 (26%) anti-MPO antibody-positive patients were vaccinated against flu and/or *S. pneumoniae* (4 in the low ceruloplasmin group and 9 in the high ceruloplasmin group). Prophylactic sulfamethoxazole trimethoprim antibiotic was taken orally by 39 (78%) anti-MPO antibody-positive patients (19 and 20 in both subgroups, respectively).

The same analyses were performed in the subgroups of 42 patients with anti-PR3 ANCA. This group consisted of 42 patients whose data are summarized in S1 Table. The median ceruloplasmin was 0.47 [0.42–0.53] g/L, allowing further division into two subgroups as done previously with the anti-MPO positive patients. Overall, there was no difference in the rate of end-stage renal disease, relapse or death between the two groups (p = 0.61, p = 0.18, p = 1, respectively). Similarly, survival and renal survival rates were comparable between the two groups (p = 0.60 and p = 0.48 respectively, S1 Fig).

In parallel, the same analyses were performed in the subpopulations of patients with MPA and GPA phenotype, according to the EMA algorithm. However, no significant differences were found regarding survival or renal survival (Tables 2 and 3, Fig 1C–1F). Using the 2022 DCVAS criterias for the classification of ANCA-associated vasculitis did not change any of our results (S2–S5 Tables).

## Discussion

To our knowledge, this is the first study associating low serum ceruloplasmin level at the diagnosis of anti-MPO ANCA-associated vasculitis with poorer survival. This prognostic impact was not found in patients with anti-PR3 ANCA-associated vasculitis.

Therefore, these findings strengthen the pathophysiological role of MPO in ANCA-associated vasculitis, supporting that a defect in MPO inhibition may be responsible for a more severe disease phenotype. Hence, the insufficient inhibition of MPO by low levels of its natural inhibitor, i.e., ceruloplasmin, is probably responsible for an insufficient inhibition of its redox cycle and of the ensuing degranulation of neutrophils and release of serum MPO, increasing the production of reactive oxygen species and tissue damage. In addition, a low level of ceruloplasmin would facilitate MPO—ANCA anti-MPO interaction and thus be responsible for more severe tissue damage. Low serum ceruloplasmin level would as well lift the inhibitory effect of HClO, produced by MPO, on α1 anti-trypsin, resulting in a decrease of its anti-inflammatory effects and an increase in serum PR3 level and consequently in tissue damage. Our data confirm that low ceruloplasmin levels are associated with worse outcome in patients with anti-MPO vasculitis. However, the prognostic impact of ceruloplasmin in the subpopulation of patients with anti-PR3 ANCA was not demonstrated.

Furthermore, results were not significant in the subgroup of patients with MPA phenotype. This may be due to a lack of power and/or the fact that the association is weaker with the phenotypic classification of ANCA-associated vasculitis. Using the 2022 DCVAS criterias for the classification of ANCA-associated vasculitis in our study did not change any of our results [8, 9]. Recent data suggest that serological status is more relevant to classify patients into homogeneous groups from an epidemiological, clinical, and therapeutic point of view [12]. Solans Loqué R. et al. showed in a retrospective study on 450 Spanish patients with ANCA-associated vasculitis that anti-MPO antibody-positive patients had twice the risk of death and almost half the risk of relapse than those with anti-PR3 antibody-positive ANCA, supporting that serological classification is better than phenotypic classification in predicting the outcome of vasculitis [13]. Lionaki E. et al. compared 3 classification systems for ANCA-associated vasculitis (Chapel Hill, EMA, and serological) in 502 patients from a US cohort. The results of this study showed a higher risk of relapse in patients with anti-PR3 ANCA than in those with anti-MPO ANCA, and this difference was not found using the other classification systems. None of the 3 systems showed superiority in terms of survival [14]. Lyons PA. et al. conducted a genomic study on 914 patients with ANCA-associated vasculitis matched to 5259 control cases. The association of genetic variants was stronger with serological than phenotypic classification [15].

To our knowledge, only two previous works have studied ceruloplasmin levels in ANCA-associated vasculitis. A pediatric study by Baskin E. et al. in 2002 involved 45 heterogeneous young patients with positive ANCA. Twenty-six of these patients exhibited perinuclear ANCA, including 16 polyarteritis nodosa, 5 renal vasculitis, 2 juvenile arthritis, 3 systemic lupus. The 18 others exhibited cytoplasmic ANCA, including 3 GPA, 3 systemic lupus, 4 juvenile arthritis, 8 small-vessel vasculitis without specificity. These authors showed that ceruloplasmin levels were increased during the active phase of the diseases, particularly in the

**Table 2. Characteristics of 46 patients with ANCA-associated vasculitis with a phenotype of granulomatosis with polyangiitis and ceruloplasmin level available at diagnosis.**

| Characteristics | (n = 46) | Low ceruloplasmin (n = 23) | High ceruloplasmin (n = 23) | P value |
|---|---|---|---|---|
| **Demographic data** | | | | |
| Age at diagnostic (years) | 64 [51–69] | 64 [49–69] | 64 [57–69] | 0.67 |
| Woman | 25 (54) | 12 (52) | 13 (57) | 0.77 |
| **ANCA type** | | | | |
| Anti-MPO | 5 (11) | 3 (13) | 2 (9) | 1 |
| Anti-PR3 | 41 (89) | 20 (87) | 21 (91) | 1 |
| **BVAS** | 20 [14–23] | 20 [16–23] | 18 [14–23] | 0.57 |
| **Characteristics of vasculitis** | | | | |
| General symptoms | 38 (83) | 20 (87) | 18 (78) | 0.70 |
| Dermatological symptoms | 8 (17) | 2 (9) | 6 (26) | 0.25 |
| Pulmonary symptoms | 30 (65) | 16 (70) | 14 (61) | 0.54 |
| ENT symptoms | 30 (65) | 15 (65) | 15 (65) | 1 |
| Ophthalmological symptoms | 6 (13) | 3 (13) | 3 (13) | 1 |
| Abdominal symptoms | 4 (9) | 3 (13) | 1 (4) | 0.61 |
| Neurological symptoms | 9 (20) | 2 (9) | 7 (30) | 0.14 |
| Cardiological symptoms | 5 (11) | 3 (13) | 2 (9) | 1 |
| Renal symptoms | 23 (50) | 12 (52) | 11 (48) | 0.77 |
| **Biological data** | | | | |
| Hematuria | 37 (82) (= 45) | 19 (83) | 18 (82) (n = 22) | 1 |
| Proteinuria | 21 (47) (n = 45) | 13 (57) | 8 (36) (n = 22) | 0.18 |
| Creatinine level (µmol/L) | 108 [65–406] (n = 45) | 197 [65–573] (n = 22) | 103 [63–177] | 0.13 |
| CRP (mg/L) | 150 [51–214] (n = 44) | 168 [17–224] (n = 21) | 117 [58–195] | 1 |
| **Treatment** | | | | |
| Induction | 44 (96) | 22 (96) | 22 (96) | 1 |
| Cyclophosphamide | 26 (57) | 16 (70) | 10 (44) | 1 |
| Rituximab | 19 (42) | 8 (35) | 11 (48) | 0.43 |
| Maintenance | 39 (89) (n = 44) | 20 (95) (n = 21) | 19 (83) | 0.35 |
| Rituximab | 28 (64) | 13 (62) | 15 (66) | 0.82 |
| Azathioprine | 13 (30) | 8 (39) | 5 (22) | 0.24 |
| Methotrexate | 4 (10) | 2 (10) | 2 (9) | 1 |
| Mycophenolate mofetil | 0 (0) | 0 (0) | 0 (0) | 1 |
| Plasma exchanges | 12 (26) | 8 (35) | 4 (17) | 0.18 |
| Bolus glucocorticoids | 42 (84) | 20 (80) | 22 (88) | 1 |
| **Relapses** | 14 (30) | 9 (39) | 5 (22) | 0.20 |
| **Deaths** | 7 (15) | 4 (17) | 3 (13) | 1 |
| **Chronic end-stage renal disease** | 4 (9) | 3 (13) | 1 (4) | 0.61 |
| **Follow-up (months)** | 57 [25–102] | 67 [24–101] | 42 [28–103] | 0.95 |

Values are given as headcount (%) or median [quartile 1-quartile 3].

PR3: proteinase 3; MPO: myeloperoxidase; BVAS: Birmingham vasculitis activity score; ENT: ear, nose and throat; CRP: C-reactive protein.

**Table 3. Characteristics of 46 patients with ANCA-associated vasculitis with a phenotype of microscopic polyangiitis and ceruloplasmin level available at diagnosis.**

| Characteristics | (n = 46) | Low ceruloplasmin (n = 23) | High ceruloplasmin (n = 23) | P value |
|---|---|---|---|---|
| **Demographic data** | | | | |
| Age at diagnostic (years) | 70 [62–73] | 71 [64–79] | 67 [62–71] | 0.15 |
| Woman | 22 (48) | 11 (48) | 11 (48) | 1 |
| **ANCA type** | | | | |
| Anti-MPO | 45 (98) | 22 (96) | 23 (100) | 1 |
| Anti-PR3 | 1 (2) | 1 (4) | 0 (0) | 1 |
| **BVAS** | 18 [14–21] | 19 [15–23] | 18 [13–21] | 0.29 |
| **Characteristics of vasculitis** | | | | |
| General symptoms | 38 (83) | 19 (83) | 19 (83) | 1 |
| Dermatological symptoms | 6 (13) | 3 (13) | 3 (13) | 1 |
| Pulmonary symptoms | 26 (57) | 12 (52) | 14 (61) | 0.56 |
| ENT symptoms | 10 (22) | 6 (26) | 4 (17) | 0.48 |
| Ophthalmological symptoms | 4 (9) | 2 (9) | 2 (9) | 1 |
| Abdominal symptoms | 2 (4) | 1 (4) | 1 (4) | 1 |
| Neurological symptoms | 12 (26) | 8 (35) | 4 (17) | 0.18 |
| Cardiological symptoms | 1 (2) | 0 (0) | 1 (4) | 1 |
| Renal symptoms | 39 (85) | 20 (87) | 19 (83) | 1 |
| **Biological data** | | | | |
| Hematuria | 36 (86) (n = 42) | 17 (77) (n = 22) | 19 (95) (n = 20) | 0.19 |
| Proteinuria | 31 (69) (n = 45) | 17 (74) | 14 (64) (n = 22) | 0.46 |
| Creatinine level (μmol/L) | 260 [136–521] | 245 [161–489] | 290 [114–606] | 0.88 |
| CRP (mg/L) | 90 [19–157] (n = 45) | 63 [14–133] | 104 [39–167] (n = 22) | 0.28 |
| **Treatment** | | | | |
| Induction | 40 (87) | 20 (87) | 20 (87) | 1 |
| Cyclophosphamide | 32 (70) | 16 (70) | 16 (70) | 1 |
| Rituximab | 9 (20) | 5 (22) | 4 (17) | 1 |
| Maintenance | 35 (78) (n = 45) | 17 (77) (n = 22) | 18 (78) | 1 |
| Rituximab | 15 (34) | 8 (37) | 7 (31) | 0.68 |
| Azathioprine | 20 (45) | 9 (41) | 11 (48) | 0.65 |
| Methotrexate | 1 (3) | 1 (5) | 0 (0) | 0.49 |
| Mycophenolate mofetil | 1 (3) | 0 (0) | 1 (4) | 1 |
| Plasma exchanges | 9 (20) | 3 (13) | 6 (26) | 0.46 |
| Bolus glucocorticoids | 40 (89) | 20 (91) | 20 (87) | 1 |
| **Relapses** | 7 (15) | 2 (9) | 5 (22) | 0.42 |
| **Deaths** | 11 (24) | 8 (35) | 3 (13) | 0.09 |
| **Chronic end-stage renal disease** | 10 (22) | 5 (22) | 5 (22) | 1 |
| **Follow-up (months)** | 38 [22–82] | 36 [19–73] | 43 [23–84] | 0.28 |

Values are given as headcount (%) or median [quartile 1-quartile 3].

PR3: proteinase 3; MPO: myeloperoxidase; BVAS: Birmingham vasculitis activity score; ENT: ear, nose and throat; CRP: C-reactive protein.

subgroup of patients with perinuclear ANCA, in comparison with their remission phase. Moreover, patients with renal involvement had significantly increased ceruloplasmin levels in the active phase of the disease, compared to those without renal involvement [16]. In contrast

to this study, our results did not show a correlation between serum levels of ceruloplasmin and serum creatinine level (p = 0.63, S3 Fig). These results may be explained by the heterogeneity of the included patients.

Ara J. *et al.* recorded ceruloplasmin levels in 21 adult patients with ANCA-associated vasculitis (14 anti-MPO ANCA, including 8 MPA and 6 renal vasculitis, and 7 anti-PR3 ANCA of GPA phenotype). They found increased levels of ceruloplasmin in the active phase of the disease compared with remission, but data was similar between the anti-MPO and anti-PR3 ANCA groups [17]. However, as an acute phase reactant, the systemic inflammatory condition related to the active phase of these diseases could explain to some extent the increase in serum ceruloplasmin levels.

Because of a persistent risk of death despite the recent progress in the treatment of ANCA-associated vasculitis, new therapies are needed, if possible non-immunosuppressive due to the increased infectious risk [18, 19]. In addition to complement system inhibition in ANCA-associated vasculitis management with avacopan [20], which nevertheless acts on the immune system, targeting of other pathophysiologic components, including biochemistry, may be assessed. Because of their major role in tissue inflammation and damage in ANCA-associated vasculitis, the products of neutrophil degranulation, namely MPO, PR3, but also other serine proteases (elastases, cathepsin G, etc.), are already the subject of scientific studies [21, 22]; these promising concepts are reinforced by our clinical data. Kusunoki Y. *et al.* found that, in the presence of an inhibitor of peptidylarginine deiminase, a component of neutrophil extracellular traps (NETs), the *in vitro* and *in vivo* levels of anti-MPO ANCA were decreased [23]. An *in vitro* study by Jerke U. *et al.* showed a decrease membrane expression of PR3, a decrease in the activation of neutrophils by anti-PR3 ANCA and a decrease in tissue damage after cathepsin C blockage, responsible for the activation of serine proteases [24]. Recently, a study by Antonelou M. *et al.* showed in an *in vivo* preclinical model that therapeutic MPO inhibition in a mouse model of ANCA-associated vasculitis reduced neutrophil degranulation, NET formation and kidney damage without increasing adaptive immune responses, suggesting that MPO inhibition may be an effective adjunctive therapy [25].

Based on our study's results, decreased ceruloplasmin levels appear to be a marker of poor prognosis in anti-MPO ANCA-associated vasculitis. Although patients with similar BVAS scores did not show a higher relapse rate, the increased number of deaths might be attributed to infectious causes. However, our sample size is too small to confirm this definitively. If this prognostic impact is confirmed, it would allow an early identification of patients at risk of unfavorable evolution in order to adapt treatments, notably by adding antibiotic prophylaxis. In addition, our data and the previous fundamental studies invite to assess the effect of a ceruloplasmin agonist. Indeed, reinforcing ceruloplasmin activity with agonist drugs could, through the increase in ferroxidase activity, simultaneously decrease the production of reactive oxygen species, limit tissue damage and insure a protection against bacterial infections [26].

The limitations of these study include its retrospective design with missing and incomplete data, and a larger patient cohort should allow a stronger statistical power.

Moreover, serum level of ceruloplasmin can be influenced by many parameters, such as systemic inflammation, hepatic and renal functions, which are frequently disturbed in ANCA-associated vasculitis. However, the monocentric study has the advantage of harmonizing patient management and this preliminary but interesting data are hopeful for this new pathophysiologic and therapeutic paradigm.

To conclude, this is, to our knowledge, the first study investigating the prognostic impact of ceruloplasmin level at ANCA-associated vasculitis diagnosis. This study suggests that a low serum ceruloplasmin level at diagnosis of anti-MPO ANCA-associated vasculitis is associated with a worse survival. These results call for further studies to understand the role of

ceruloplasmin in the pathophysiology of ANCA-associated vasculitis, as well as larger retrospective or prospective studies to confirm our findings and to pursue the research of targeting of the ceruloplasmin/MPO complex in ANCA-associated vasculitis.

## Supporting information

**S1 Fig. Survival curves and renal survival curves of patients with anti-proteinase 3 ANCA-associated vasculitis.**
(TIF)

**S2 Fig. ROC curve to determine the cut off levels of serum ceruloplasmin to discriminate survival.**
(TIF)

**S3 Fig. Scatter plot showing the relationship between serum ceruloplasmin and creatinine levels.**
(TIF)

**S1 Table. Characteristics of 42 patients with anti-proteinase 3 ANCA-associated vasculitis with ceruloplasmin level available at diagnosis.** Values are given as headcount (%) or median [quartile 1-quartile 3]. GPA: granulomatosis with polyangiitis; MPA: microscopic polyangiitis; BVAS: Birmingham vasculitis activity score; ENT: ear, nose and throat; CRP: C-reactive protein.
(DOCX)

**S2 Table. Characteristics of 48 patients with anti-MPO ANCA-associated vasculitis with ceruloplasmin level available at diagnosis, using the 2022 DCVAS criterias for the classification of ANCA-associated vasculitis.** Values are given as headcount (%) or median [quartile 1-quartile 3]. GPA: granulomatosis with polyangiitis; MPA: microscopic polyangiitis; PR3: proteinase 3; MPO: myeloperoxidase; BVAS: Birmingham vasculitis activity score; ENT: ear, nose and throat; CRP: C-reactive protein.
(DOCX)

**S3 Table. Characteristics of 45 patients with ANCA-associated vasculitis with a phenotype of granulomatosis with polyangiitis and ceruloplasmin level available at diagnosis, using the 2022 DCVAS criterias for the classification of ANCA-associated vasculitis.** Values are given as headcount (%) or median [quartile 1-quartile 3]. PR3: proteinase 3; MPO: myeloperoxidase; BVAS: Birmingham vasculitis activity score; ENT: ear, nose and throat; CRP: C-reactive protein.
(DOCX)

**S4 Table. Characteristics of 45 patients with ANCA-associated vasculitis with a phenotype of microscopic polyangiitis and ceruloplasmin level available at diagnosis, using the 2022 DCVAS criterias for the classification of ANCA-associated vasculitis.** Values are given as headcount (%) or median [quartile 1-quartile 3]. PR3: proteinase 3; MPO: myeloperoxidase; BVAS: Birmingham vasculitis activity score; ENT: ear, nose and throat; CRP: C-reactive protein.
(DOCX)

**S5 Table. Characteristics of 42 patients with anti-PR3 ANCA-associated vasculitis with ceruloplasmin level available at diagnosis, using the 2022 DCVAS criterias for the classification of ANCA-associated vasculitis.** Values are given as headcount (%) or median [quartile 1-quartile 3]. GPA: granulomatosis with polyangiitis; MPA: microscopic polyangiitis; PR3:

proteinase 3; MPO: myeloperoxidase; BVAS: Birmingham vasculitis activity score; ENT: ear, nose and throat; CRP: C-reactive protein.
(DOCX)

## Author Contributions

**Conceptualization:** Samuel Deshayes.

**Data curation:** Louis Camboulive.

**Formal analysis:** Louis Camboulive.

**Methodology:** Louis Camboulive, Samuel Deshayes.

**Supervision:** Samuel Deshayes.

**Validation:** Frédérique Grandhomme, Nicolas Martin Silva, Achille Aouba, Samuel Deshayes.

**Visualization:** Kathy Khoy, Delphine Mariotte, Thierry Lobbedez, Anaël Dumont, Alexandre Nguyen, Hubert de Boysson.

**Writing – original draft:** Louis Camboulive.

**Writing – review & editing:** Louis Camboulive, Samuel Deshayes.

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
