## [Decision Letter · Decision Letter 0]

29 Jul 2024

PONE-D-24-07849Clinical impact of ceruloplasmin levels at ANCA-associated vasculitis diagnosisPLOS ONE

Dear Dr. CAMBOULIVE,

Thank you for submitting your manuscript to PLOS ONE. After careful consideration, we feel that it has merit but does not fully meet PLOS ONE’s publication criteria as it currently stands. Therefore, we invite you to submit a revised version of the manuscript that addresses the points raised during the review process.

 Specifically, our reviewers found some interests in this study, but pointed out a number of comments and suggestions that require improvement and amendment. I ask the authors to fully respond to all comments made by reviewers in the revised version. 

We look forward to receiving your revised manuscript.

Kind regards,

Masataka Kuwana, MD, PhD

Academic Editor

PLOS ONE

Reviewers' comments:

Reviewer's Responses to Questions

**Comments to the Author**

1. Is the manuscript technically sound, and do the data support the conclusions?

Reviewer #1: Yes

Reviewer #2: Yes

2. Has the statistical analysis been performed appropriately and rigorously? 

Reviewer #1: Yes

Reviewer #2: Yes

3. Have the authors made all data underlying the findings in their manuscript fully available?

Reviewer #1: Yes

Reviewer #2: Yes

4. Is the manuscript presented in an intelligible fashion and written in standard English?

Reviewer #1: Yes

Reviewer #2: Yes

5. Review Comments to the Author

Reviewer #1: This retrospective monocentric study conducted at Caen University Hospital aims to evaluate the prognostic impact of serum ceruloplasmin levels at the diagnosis of AAV. The study involved 92 patients with anti-MPO or anti-PR3 antibodies. The results suggest that a low serum level of ceruloplasmin at diagnosis is associated with worse survival in anti-MPO AAV patients but not in anti-PR3 AAV patients.

Previous studies have not thoroughly examined the prognostic implications of ceruloplasmin levels in AAV, thus the findings are significant as they highlight a potential biomarker for prognosis in anti-MPO AAV, which could influence treatment decisions and patient management. However, some conrcenrs should be addressed.

1. the authors should discuss more deeply the pathological significance of MPO in AAV and the significance of ceruloplasmin as a defense mechanism against it in the introduction.

2. please consider providing more detailed demographic data to better understand the patient population in the results, such as initial PSL dose, use of immunosuppressive agents etc.

3. the authors should consider addressing the potential confounding factors that might influence ceruloplasmin levels, more thoroughly in the discussion.

4. the authors should consider why only survival is different, whereas there are no differences in the baseline clinical characteristics and the recurrence rates of patients with and without elevated ceruloplasmin levels.

Reviewer #2: The authors retrospectively examined the serum levels of ceruloplasmin in 50 patients with anti-MPO antibodies-positive AAV, and demonstrated that the mortality was significantly higher in patients with low serum ceruloplasmin levels as compared to those with high serum levels of ceruloplasmin, albeit no significant differences in relapse rate or renal failure was observed between the two groups. No such difference was observed in patients with anti-PR3 antibodies-positive AAV,

Minor points to be addressed.

(1) Please specify the race of patients with ANCA-associated vasculitis, since anti-MPO-antibodies are prevalent in GPA in Asian countries including Japan (Rheumatology 50:1916, 2011; J Rheumatol 44: 216, 2017; Arthritis Res Ther 16: R101, 2014 et al).

(2) Please determine the cut off levels of serum ceruloplasmin to discriminate the survival by ROC curve analysis.

(3) It should be mentioned whether serum levels of ceruloplasmin are determined genetically by polymorphism or not.

6. PLOS authors have the option to publish the peer review history of their article (what does this mean?). If published, this will include your full peer review and any attached files.

Reviewer #1: **Yes: **Ryu Watanabe

Reviewer #2: No

---

## [Author Response · Author response to Decision Letter 0]

4 Sep 2024

Dear Reviewer 1, thank you for your valuable feedback. Please find our responses below. 

- To address this first concern, here is the modification we have brough to the introduction “MPO is a key component of neutrophil granules, capable of producing hypochlorous acid (HClO), which contributes to tissue destruction and inhibits α1 anti-trypsin, a plasma inhibitor of PR3. In ANCA-associated vasculitis, MPO is primarily expressed on the surface of activated neutrophils, where anti-MPO antibodies bind to it, triggering the release of reactive oxygen species. This process leads to the formation of neutrophil extracellular traps (NETs), which have cytotoxic properties that cause inflammation and vascular injury.” 

We also added the following phrase in the introduction “inadequate inhibition of MPO by low ceruloplasmin levels may contribute to a more severe phenotype in ANCA-associated vasculitis by enhancing oxidative stress and tissue damage through increased MPO activity and interaction with ANCA anti-MPO.”

- We have taken your advice into consideration and modified our results accordingly. 

- Regarding the cofounding confounding factors that might influence ceruloplasmin levels, we have added the following details to the manuscript: “Moreover, serum level of ceruloplasmin can be influenced the following parameters: systemic inflammation, hepatic and renal functions (which are frequently disturbed in ANCA-associated vasculitis). Ceruloplasmin serum levels are more often increased during systemic inflammation and impaired renal function, whereas they are decreased in the case of liver failure.”

- We thank you for this highly relevant remark. The low number of deaths in our study makes reliable analysis difficult. The patients who died had not experienced a relapse prior to their death (except for 1 patient), which prevents them from relapsing. 

We have made some modifications to the manuscript (in the discussion) in regards to your comments. 

“Based on our study's results, decreased ceruloplasmin levels appear to be a marker of poor prognosis in anti-MPO ANCA-associated vasculitis. Although patients with similar BVAS scores did not show a higher relapse rate, the increased number of deaths might be attributed to infectious causes. However, our sample size is too small to confirm this definitively.”

Dear reviewer 2, we thank you for your comments, here are our responses: 

- In France, we are not allowed to collect the race of the patients. However, the recruitment is taking place in France, where the majority of GPA cases are found, both nationally and in our center. (https://link.springer.com/article/10.1186/s41927-024-00385-8), (https://pubmed.ncbi.nlm.nih.gov/30805643/)

- Following your review, we have added the cut off levels of serum ceruloplasmin to discriminate the survival by ROC curve analysis. The ROC curve is added as the S2 Fig.

- There do seem to be polymorphisms that influence ceruloplasmin levels, but these were not investigated in our study.

---

## [Decision Letter · Decision Letter 1]

24 Sep 2024

Clinical impact of ceruloplasmin levels at ANCA-associated vasculitis diagnosis

PONE-D-24-07849R1

Dear Dr. CAMBOULIVE,

We’re pleased to inform you that your manuscript has been judged scientifically suitable for publication and will be formally accepted for publication once it meets all outstanding technical requirements.

Kind regards,

Masataka Kuwana, MD, PhD

Academic Editor

PLOS ONE

Additional Editor Comments (optional):

Reviewers' comments:

Reviewer's Responses to Questions

**Comments to the Author**

1. If the authors have adequately addressed your comments raised in a previous round of review and you feel that this manuscript is now acceptable for publication, you may indicate that here to bypass the “Comments to the Author” section, enter your conflict of interest statement in the “Confidential to Editor” section, and submit your "Accept" recommendation.

Reviewer #1: All comments have been addressed

Reviewer #2: All comments have been addressed

2. Is the manuscript technically sound, and do the data support the conclusions?

Reviewer #1: Yes

Reviewer #2: Yes

3. Has the statistical analysis been performed appropriately and rigorously? 

Reviewer #1: Yes

Reviewer #2: Yes

4. Have the authors made all data underlying the findings in their manuscript fully available?

Reviewer #1: Yes

Reviewer #2: Yes

5. Is the manuscript presented in an intelligible fashion and written in standard English?

Reviewer #1: Yes

Reviewer #2: Yes

6. Review Comments to the Author

Reviewer #1: The authors revised the manuscript based on the reviewers' comments. I think the manuscript is now acceptable to this journal.

Reviewer #2: (No Response)

7. PLOS authors have the option to publish the peer review history of their article (what does this mean?). If published, this will include your full peer review and any attached files.

Reviewer #1: **Yes: **Ryu Watanabe

Reviewer #2: No

---

## [Editor Report · Acceptance letter]

29 Sep 2024

PONE-D-24-07849R1 

PLOS ONE

Dear Dr. CAMBOULIVE, 

I'm pleased to inform you that your manuscript has been deemed suitable for publication in PLOS ONE. Congratulations! Your manuscript is now being handed over to our production team.

Kind regards, 

on behalf of

Prof. Masataka Kuwana 

Academic Editor

PLOS ONE